# Functional Magnetic Resonance Imaging and Diffusion Tensor Imaging-Tractography in Resective Brain Surgery: Lesion Coverage Strategies and Patient Outcomes

**DOI:** 10.3390/brainsci13111574

**Published:** 2023-11-09

**Authors:** Vasileios Kokkinos, Athanasios Chatzisotiriou, Ioannis Seimenis

**Affiliations:** 1Department of Neurosurgery, Massachusetts General Hospital, Boston, MA 02114, USA; 2Harvard Medical School, Boston, MA 02215, USA; 3Department of Neurosurgery, St. Luke’s Hospital, 55236 Thessaloniki, Greece; achatzisot@gmail.com; 4Department of Medicine, School of Health Sciences, Democritus University of Thrace, 387479 Alexandroupolis, Greece; iseimen@med.uoa.gr

**Keywords:** fMRI, DTI-tractography, neuronavigation, brain tumor surgery, presurgical evaluation

## Abstract

Diffusion tensor imaging (DTI)-tractography and functional magnetic resonance imaging (fMRI) have dynamically entered the presurgical evaluation context of brain surgery during the past decades, providing novel perspectives in surgical planning and lesion access approaches. However, their application in the presurgical setting requires significant time and effort and increased costs, thereby raising questions regarding efficiency and best use. In this work, we set out to evaluate DTI-tractography and combined fMRI/DTI-tractography during intra-operative neuronavigation in resective brain surgery using lesion-related preoperative neurological deficit (PND) outcomes as metrics. We retrospectively reviewed medical records of 252 consecutive patients admitted for brain surgery. Standard anatomical neuroimaging protocols were performed in 127 patients, 69 patients had additional DTI-tractography, and 56 had combined DTI-tractography/fMRI. fMRI procedures involved language, motor, somatic sensory, sensorimotor and visual mapping. DTI-tractography involved fiber tracking of the motor, sensory, language and visual pathways. At 1 month postoperatively, DTI-tractography patients were more likely to present either improvement or preservation of PNDs (*p* = 0.004 and *p* = 0.007, respectively). At 6 months, combined DTI-tractography/fMRI patients were more likely to experience complete PND resolution (*p* < 0.001). Low-grade lesion patients (N = 102) with combined DTI-tractography/fMRI were more likely to experience complete resolution of PNDs at 1 and 6 months (*p* = 0.001 and *p* < 0.001, respectively). High-grade lesion patients (N = 140) with combined DTI-tractography/fMRI were more likely to have PNDs resolved at 6 months (*p* = 0.005). Patients with motor symptoms (N = 80) were more likely to experience complete remission of PNDs at 6 months with DTI-tractography or combined DTI-tractography/fMRI (*p* = 0.008 and *p* = 0.004, respectively), without significant difference between the two imaging protocols (*p* = 1). Patients with sensory symptoms (N = 44) were more likely to experience complete PND remission at 6 months with combined DTI-tractography/fMRI (*p* = 0.004). The intraoperative neuroimaging modality did not have a significant effect in patients with preoperative seizures (N = 47). Lack of PND worsening was observed at 6 month follow-up in patients with combined DTI-tractography/fMRI. Our results strongly support the combined use of DTI-tractography and fMRI in patients undergoing resective brain surgery for improving their postoperative clinical profile.

## 1. Introduction

The surgical treatment for a variety of brain lesions and tumors still comprises craniotomy and resection [1]. Resective surgery aims to relieve patients from the associated risks in cases of benign lesions and low-grade tumors [2,3,4] and to extend survival and improve the patients’ quality of life when high-grade life-threatening tumors are involved [5,6,7,8,9,10,11]. However, although neurosurgical techniques have been constantly improving [12,13], the inherent space-occupying and infiltrating nature of many brain tumors keeps the risk of introducing new neurological deficits or worsening of existing ones by surgical maneuvers and aggressive resection significantly high [14,15]. The challenges are particularly increased in high-grade gliomas residing adjacent to or infiltrating eloquent cortex, where patients are faced with higher morbidity [6,10] and resections are often limited and incomplete to preserve their quality of life [16,17,18].

To address these challenges of resective brain tumor surgery, a variety of neuroimaging techniques have been introduced in the presurgical and intraoperative workflow, with functional magnetic resonance imaging (fMRI) and diffusion tensor imaging tractography (DTI-tractography) being the predominant ones. fMRI is a scientifically and clinically non-invasive neuroimaging technique for highlighting and localizing functional representation in grey matter regions of the brain [19,20]. Moreover, and for the past two decades, fMRI has been established as a key part of presurgical evaluations for brain surgery [21,22,23,24,25,26,27,28], as it has demonstrated reliable mapping of all key functions subserved by eloquent cortex, such as primary motor function [29], for both fine [30,31] and coarse movements [32,33], primary somatic sensation [30,34,35], primary vision [36,37], as well as primary expressive and receptive language functions [38,39,40]. Despite the challenges mainly stemming from technical and performance aspects [28,41,42], fMRI remains an integral part of presurgical evaluation, as it has been associated with significant reduction of morbidity and mortality, as well as higher postoperative quality of life [43,44,45].

Another popular neuroimaging technique employed to support resective brain tumor surgeries is DTI-tractography. Tractography is the three-dimensional representation of the brain’s white matter fiber tracts, most often derived from the degree and directionality of water diffusion measured by diffusion tensor imaging (DTI) [46,47] and implemented by seeding regions of interest (ROI) [46,48,49]. Fiber tracking has been extensively used in the clinical setting to identify white matter bundles in the vicinity of brain lesions [50,51,52,53,54,55,56,57,58]. Despite the persistence of technical problems such as that of fiber crossing [59,60,61,62,63] that tend to worsen in the presence of peri-lesional edema [64,65], DTI-tractography has been successfully used in presurgical evaluation and intraoperative workflows and has been associated with low complication rates and preservation or improvement of patients’ quality of life [66,67,68].

The complementary character of fMRI and DTI-tractography in providing a complete map of the grey matter/white matter integration that underlies function and structure in the brain has resulted in the use of both modalities during presurgical evaluations [69,70,71]. More than often, fMRI clusters have been used as seeding ROIs for fiber tracking [46,71,72,73,74,75,76], thereby enhancing the specificity of DTI-tractography for the corresponding fMRI-mapped function. However, both the technical short-comings of fMRI, such as task-based sensitivity, statistical thresholding, feasibility issues, etc. [42,77], and the requirement for prolonged MR scanning in a stressful and time-demanding presurgical setting often raise the question of efficiency in relevant procedures.

In order to identify features that denote efficient use of these presurgical functional neuroimaging assets versus using structural imaging alone, we set out to assess the added value of using DTI-tractography and combined fMRI/DTI-tractography, utilizing patient outcomes in terms of lesion-related neurological deficits as metrics. We are also presenting the location-specific functional coverage strategy we developed and used to surround the lesion in order to optimize the presurgical planning and the intraoperative lesion access trajectory.

## 2. Materials and Methods

### 2.1. Patients

Our retrospective study used a pool of two hundred and fifty-two (252) consecutive patients admitted for elective brain surgery at St. Luke’s Hospital (Thessaloniki, Greece) between November 2009 and August 2017. All resections and intraoperative decisions were performed by a single experienced neurosurgeon (AC). Postoperative histopathological evaluations were based on the World Health Organization (WHO) 2007 classification system.

### 2.2. Anatomical MRI

MRI was performed with a 1.5 T Magnetom Avanto scanner (Siemens AG, Germany) equipped with a Tim 32 RF system and a Q-engine gradient system (33 mT/m per axis and 57 mT/m all axes combined). A 12-element dedicated head coil was used for signal reception. All patients underwent presurgical morphological/anatomical neuroimaging with a standard brain lesion/tumor protocol including a 3D T1-weighted sequence, run pre- and post-Gadolinium, and a fluid attenuated inversion recovery (FLAIR) sequence. All sequences were prescribed in the sagittal plane and images were acquired with 1 mm isotropic spatial resolution [78]. The standard neuroimaging protocol was performed in 127 patients, while 69 patients had additional DTI-tractography and 56 had both DTI-tractography and fMRI during their presurgical evaluation phase. This categorization was the result of the progressive introduction of pre-surgical neuroimaging modalities into the clinical workflow, rather than a criteria-based selection.

### 2.3. Functional MRI and Tasks

All patients, after admission, went through a thorough screening procedure to assess their ability to perform each fMRI task. More specifically, outside the MR scanner, they were presented with samples of the fMRI tasks they were referred for and were asked to perform each task in front of the functional neuroimager (VK). The screening language fMRI tasks contained different material than the tasks used during the functional scan but had the same duration and structure. The screening sensory, motor and sensorimotor tasks were exactly the same as the ones used during the functional scan. For the language fMRI tasks, a performance comprising incomplete sentences/words and/or systematic failure to initiate sentences/words promptly in >50% of the presented material, failure to read the sentences presented in >50% of the material, or recall <50% of key items of the listening comprehension material, accompanied by excessive gaps of silence and/or sound prolongation, would disqualify patients from undergoing the task in the scanner [42]. For the motor and sensorimotor fMRI tasks, a performance that sustained motion for <50% of the active block interval would disqualify patients. There were no screening criteria for the sensory fMRI tasks, as they required no patient effort.

A gradient echo T2*-weighted single-shot echo-planar imaging (EPI) pulse sequence (echo time [TE]: 45 msec; repetition time [TR]: 4000 msec; flip angle: 90°; field of view: 240 cm × 240 cm; acquisition matrix: 96 × 96) was employed for fMRI, yielding 45 axial slices of 2.5 mm thickness each and 0.3 mm inter-slice distance. Language and visual tasks were designed with alternating 10 active periods and 11 baseline periods, with each active and baseline block lasting 20 s (5 volumes each), and a total task time of 7 min (total 105 volumes). Motor [31,32,33], somatosensory [30], mixed sensorimotor [32,79] and visual [37,80] tasks were designed with alternating 5 active periods and 6 baseline periods, with each active and baseline block lasting 20 s (5 volumes each), and a total task time of 3 min and 40 s (total 55 volumes). In all tasks, 2 dummy volume scans were added at the beginning of each sequence (8 s) to allow for T1 saturation; these volumes were discarded from further analysis. All instructions and task information were provided visually through a computer/projector/back-projection screen system, with the exception of the listening comprehension task that was provided auditorily through MR-compatible headphones.

The fMRI analysis workflow for each separate task included a standard image preprocessing stage, which comprised realignment to the mean EPI image (B-Spline interpolation at 2.5 mm), normalization in MNI space and smoothing with a Gaussian kernel of 8 mm. In turn, a general linear model (GLM) was generated using SPM8 (Wellcome Trust Centre for Imaging Neuroscience, London, UK; http://www.fl.ion.ucl.ac.uk/spm, accessed on 1 November 2009). The stimulus intervals in the GLM were represented as blocks for all fMRI task analysis (block design). At the level of individual participants, maps of t-plus contrast for the effect of the performed functional blocks were obtained at a significance threshold of *p* < 0.05.

#### 2.3.1. Language Tasks

The aim of language fMRI is the mapping of the receptive (Wernicke’s) and expressive (Broca’s) language regions. Functional mapping of expressive and receptive language areas included four tasks: a verbal fluency task, a semantic description task, a reading comprehension task and a listening comprehension task (Figure 1A–D). The purpose of the verbal fluency task was to activate expressive language regions associated with vocabulary use. The purpose of the semantic description task was to activate language areas associated with sentence formation and semantic integration skills. The purpose of the reading comprehension task was to activate regions of the language network associated with reading skills. Finally, the purpose of the listening comprehension task was to activate receptive language regions associated with understanding auditorily provided speech (more detailed descriptions can be found in [42]). All verbal fluency, semantic description and reading comprehension tasks were performed in the maternal languages of our patients and included Greek, Albanian, Slavic, Russian, Egyptian and Armenian. Listening comprehension was available only in the Greek language, as it required pre-recorded material, and was offered only to Greek native speakers. The efficiency of these language fMRI tasks to activate the expressive inferior frontal and receptive posterior parieto-temporal regions has been shown in a previous study of ours [42]. More specifically, the verbal fluency task was sensitive for expressive language, although less sensitive for receptive language skills. The semantic description task was sensitive for both expressive and receptive language skills. The reading comprehension task was less sensitive for expressive language skills but typically sensitive for receptive language. The listening comprehension task was also less sensitive for expressive language but was highly sensitive for receptive language skills.

The verbal fluency, semantic description and reading comprehension tasks shared the same non-linguistic visuo-motor exploration baseline task (Figure 1A–C). Each slide contained 4 arrays of randomly aligned symbols from the ASCII table of characters, and patients were asked to visually search and fixate their gaze on the smiley characters (☺ and ☻). The listening comprehension task had a baseline comprising the same narrative delivered during the active phases but digitally inverted. The purpose was to isolate the comprehension component in the contrast and avoid known bilateral activations owing to the auditory nature of the task.

#### 2.3.2. Motor Tasks

The aim of motor fMRI is to map the motor functions across the primary motor cortex. Functional motor mapping of the upper extremities was performed by voluntary sequential extension and flexion of the wrist and fingers during the active phases. Functional motor mapping of the lower extremities was performed by voluntary extension and flexion of the foot under manual resistance applied by the functional neuroimager (VK) following visually provided instructions (Figure 1E). The respective limbs remained relaxed and immobile during the baseline phases [31,32,33].

#### 2.3.3. Somatosensory Tasks

The aim of somatosensory fMRI is to identify the sensorimotor representation across the primary somatosensory cortex. Functional sensory mapping of the upper extremities was performed by passive superficial sensory manipulation of the dorsal hand (opisthenar) by the functional neuroimager (VK) following visually provided instructions (Figure 1E). Functional sensory mapping of the lower extremities was performed by passive superficial sensory manipulation of the foot dorsal region (dorsum) [30].

#### 2.3.4. Sensorimotor Tasks

Sensorimotor fMRI tasks are mapping simultaneously motor and somatosensory functions, resulting in the activation of both primary motor and primary somatosensory cortices. This is because separating motion from sensation during the task performance is difficult or impractical [32,79]. Mixed peri-oral motor and sensory mapping was performed by voluntary consecutive circular movements of the tongue between the teeth and the lips by the patient following visually provided instructions (Figure 1E). The tongue remained relaxed during the baseline phases.

#### 2.3.5. Visual Tasks

The aim of visual fMRI in the clinical setting is to highlight the distribution of visual processing in the primary visual cortex and visual association regions. Functional mapping of vision was performed by passive exposure to high-definition colorful landscapes, during which the patient was encouraged to visually explore the scenery, alternating with black-screen baseline intervals during which the patients were asked to focus their gaze on a cross in the center of the screen (Figure 1F). The scanner room lights remained off throughout the process [37,80].

### 2.4. DTI-Tractography and Tracts

The acquisition of DTI data involved an axial single-shot spin-echo EPI sequence with 6 diffusion encoding directions [81,82] and the following parameters: TR: 8000 ms, TE: 94 ms, flip angle: 90°, field of view: 220 cm × 220 cm; acquisition matrix: 96 × 96, voxel size: 2 × 2 × 4 mm^3^, b values: 0 and 1000 s/mm^2^, 50 slices, total scan time: 4 min. Following acquisition, diffusion imaging data were motion-corrected relative to the b = 0 image series of each subject, and eddy current correction was performed by the acquisition suite (Syngo MR Neuro3D Engine VB10, Siemens AG, Erlagen, Germany). In turn, and for each voxel, the diffusion tensor was estimated, tensor decomposition was performed, image fractional anisotropy (FA) of diffusivity was calculated for b = 0 and 1000 s/mm^2^ and FA maps were generated. After image preprocessing, deterministic tractography was performed with the following stopping criteria: FA < 0.25 and >35° angle between two subsequent directions. Multiple ROIs were used with a logical AND to highlight each tract [48,83], as described below, and a logical NOT was often applied to exclude fiber tracts not anatomically consistent with the tract bundle of interest, as indicated below. No spatial masking approaches were used. Four (4) were the main tracts used to assist the intraoperative neuronavigation process: 1. the cortico-spinal tract (the motor pathway), 2. the spino-thalamic and thalamo-cortical tracts (the sensory pathway), 3. the arcuate fasciculus (the language network) and 4. the optic radiation (the posterior optic pathway). Although it is currently acknowledged that more fiber pathways are involved in the integration of brain function, at the time of introducing DTI-tractography in our center’s workflow, these four tracts were the best described in the published literature, allowing for confident replication of the fiber-tracking process.

#### 2.4.1. Cortico-Spinal Tract (The Motor Pathway)

The cortico-spinal tracts are descending projection tracts that connect the primary motor cortex to the motor neurons of the spinal cord. To delineate the cortico-spinal tracts that form the motor pathway, two ROIs on axial plane were used: 1. within brainstem space, over the ipsilateral cerebral peduncle running in the superior-inferior direction (blue-coded), residing rostral and lateral in the anterior pons [53,57], and 2. within neocortical space, over the precentral gyrus. When the hemispheric ROI was distorted by tumor or peri-lesional edema, a ROI within the posterior limb of the internal capsule was additionally used [58].

#### 2.4.2. Spino-Thalamic and Thalamo-Cortical Tracts (The Sensory Pathway)

The somatosensory pathway comprises ascending projection tracts that connect the sensory neurons of the spinal cord to the primary somatosensory cortex through the thalamus. To delineate the spino-thalamic and thalamo-cortical tracts that constitute the ascending somatosensory pathway, two ROIs on axial plane were used: 1. within brainstem space, over the ipsilateral medial lemniscus running in the superior-inferior direction (blue-coded), residing caudal and mesial in the posterior pons [54,57], and 2. within neocortical space, over the postcentral gyrus. In cases where the postcentral ROI was distorted by tumor or peri-lesional edema, an additional ROI within the posterior limb of the internal capsule was used.

#### 2.4.3. Arcuate Fasciculus (The Language Network)

The arcuate fasciculus is a longitudinal tract connecting the posterior temporal receptive language region (Wernicke’s) with the inferior frontal expressive language region (Broca’s). To delineate the arcuate fasciculus, an ROI in the coronal plane over tracts running in the anterior-posterior direction (green-coded) in supra-Sylvian space, at the level of the posterior limb of the internal capsule and the rostral corpus callosum and lateral to the corona radiata, was used [53,55]. Fibers of the superior longitudinal fasciculus extending in basal frontal/prefrontal space were removed by logical NOT.

#### 2.4.4. Optic Radiation (The Visual Pathway)

The optic radiation constitutes the posterior part of the visual pathway, originating in the lateral geniculate nucleus (LGN) of the thalamus, looping within anterior temporal lobe space (Meyer’s loop) over the lateral ventricle’s temporal horn and terminating in the calcarine fissure residing in the mesial occipital lobe. To delineate the optic radiation, two ROIs were used: 1. within thalamic space, over a bundle of fibers running in the left-to-right direction (red-coded) at the level of the LGN in the sagittal plane, and 2. within temporal lobe space, over tracts running in the anterior-posterior direction (green-coded) adjacent to the lateral wall of the lateral ventricle’s occipital horn [56,57] in the coronal plane. Fibers of the inferior occipito-frontal fasciculus extending in anterior temporal/prefrontal space beyond Meyer’s loop were removed by logical NOT.

### 2.5. Strategy and Rationale of Functional and Structural Peri-Lesional Coverage

The aim of our lesion coverage strategy by means of fMRI and DTI-tractography mapping was two-fold: 1. Delineate the position of primary eloquent cortices and tracts underlying basic brain functions in the immediate vicinity of the lesion. 2. Identify the optimal lesion access trajectory that invasively reaches the lesion without damaging the surrounding functional regions or white matter tracts. In order to minimize redundancy in exams performed and respect the allocated scanner time, patients did not perform all fMRI tasks or have all their tracts highlighted by DTI-tractography. Instead, we employed a personalized and region-specific approach that included DTI-tractography maps alone or in combination with fMRI and are outlined in Table 1.

#### 2.5.1. Precentral Lesions

For lesions residing in the posterior parts of the superior and middle frontal gyrus (Brodmann areas 8 and 6, respectively), anterior to the precentral gyrus, the main surgical concerns regarded the establishment of the proximity to the motor and the language tracts. For such lesions, the mapping process included fMRI of the primary contralateral hand and foot representations and DTI-tractography of the ipsilateral cortico-spinal tracts and the arcuate fasciculus (Figure 2A,B).

#### 2.5.2. Central Lesions

For lesions residing within and/or adjacent to the precentral/postcentral gyri (Brodmann areas 1, 2, 3, and 4), the main surgical concerns were the integrity of both eloquent cortices and the integrity of their respective tracts. For such lesions, the mapping process included motor and sensory fMRI of the contralateral hand/foot and DTI-tractography of the ipsilateral cortico-spinal and the combined spino-thalamic/thalamo-cortical tracts (Figure 2C,D).

#### 2.5.3. Inferior Frontal Lesions

For lesions residing in the inferior frontal gyrus (Brodmann areas 47, 45 and 44) and/or the middle frontal gyrus anteriorly (Brodmann area 46), especially in the language-dominant hemisphere, the main surgical concerns were the proximity to Broca’s area, the proximity to the inferior precentral gyrus and the integrity of the arcuate fasciculus. For such lesions, the mapping process included language fMRI (accompanied by peri-oral motor fMRI, depending on the posterior extent of the lesion, and DTI-tractography of the ipsilateral arcuate and cortico-spinal tracts (Figure 3A,B).

#### 2.5.4. Inferior Parietal/Posterior Temporal Lesions

For lesions residing in the inferior parietal lobule/posterior temporal region (Brodmann areas 39, 40, 41, 42 and 22), especially in the language-dominant hemisphere, the main surgical concerns were the proximity to Wernicke’s area, the proximity to the inferior post-central gyrus and the integrity of the arcuate fasciculus. For such lesions, the mapping process included language fMRI and DTI-tractography of the ipsilateral arcuate, the combined spino-thalamic/thalamo-cortical tracts (optionally accompanied by the cortico-spinal tracts, depending on the anterior extent of the lesion) and the optic radiation (Figure 3C,D).

#### 2.5.5. Temporo-Occipital Lesions

For lesions residing in the temporo-occipital regions (Brodmann areas 19, 30 and 37), the main surgical concerns were the proximity to primary visual and receptive language areas, the proximity to the primary somatosensory cortex and the integrity of the optic radiation. For such lesions, the mapping process included visual fMRI (and language fMRI depending on the extent of the lesion), as well as DTI-tractography of the arcuate fasciculus, the combined spino-thalamic/thalamo-cortical tracts and the optic radiation (Figure 4A,B).

#### 2.5.6. Anterior Temporal Lesions

For lesions residing in the anterior part of the temporal lobe, the main surgical concern was the proximity to the optic radiation’s Meyer’s loop. For such lesions, the mapping process included DTI-tractography of the optic radiation.

#### 2.5.7. Anterior Frontal Lesions

For lesions residing in the anterior part of the frontal lobe, the main surgical concern was the proximity to the expressive language areas (Brodmann areas 44, 45) and the integrity of the anterior part of the arcuate fasciculus. For such lesions, the mapping process included language fMRI and DTI-tractography of the arcuate fasciculus. For lesions closer to the orbit, the base of the frontal lobe, the mesial wall and the anterior middle and superior frontal gyrus, the language fMRI was omitted.

#### 2.5.8. Superior Parietal Lesions

For lesions in the superior part of the parietal lobe (Brodmann areas 5 and 7), the main surgical concern was the proximity to the primary somatosensory cortex and the respective thalamo-cortical tracts. For such lesions, the mapping process included sensory fMRI and DTI-tractography of the combined spino-thalamic/thalamo-cortical tracts. For lesions closer to the posterior mesial and lateral parietal lobe, the sensory fMRI was omitted.

#### 2.5.9. Occipital Lesions

For lesions in the occipital lobe (Brodmann areas 17, 18 and 19), the main surgical concern was the proximity of the lesion to primary visual areas and the integrity of the optic radiation. For such lesions, the mapping process included visual fMRI and DTI-tractography of the optic radiation. For lesions closer to the lateral surface of the occipital lobe, the visual fMRI was omitted.

#### 2.5.10. Lesions Extending in Two Lobes

For lesions of extended spatial distribution, the main surgical concerns were the proximity to surrounding eloquent areas and the integrity of tracts in the immediate peri-lesional vicinity. For such lesions, the mapping process included fMRI and DTI-tractography to highlight the location of all the relevant and critical regions/tracts surrounding the lesion (Figure 4C,D).

### 2.6. Intraoperative Procedure

All fMRI and DTI-tractography extracts were overlaid on post-Gd T1-weighted MR images. Anatomical, structural and functional data were fused in a common 3D coordinate space, with the post-Gd T1-weighted image as reference, using the neuronavigation system (Sonowand, Trondheim, Norway; BrainLab AG, Munich, Germany).

All patients underwent brain surgery under general anesthesia. Awake craniotomy procedures were not performed. After induction of general anesthesia, the patient’s head was fixed to the surgical table through a Mayfield 3 pin head clamp and a neuronavigation reference device was in turn firmly attached to the Mayfield. Co-registration of the patient’s physical space to the MRI’s digital space was performed either by fiducial markers (placed on the patient’s head before the MRI procedure in a bilateral and asymmetric fashion) or by means of facial surface-point matching using a laser probe. A co-registration accuracy of ≤1.2 mm at all markers or points was acceptable for the procedure. Manual verification of accuracy was performed using a non-sterile hand-help neuronavigation pointer placed over distinctive anatomical landmarks of the patient’s head and/or moved over the patient’s scalp. Upon achievement of satisfactory accuracy, the patient’s head was locally shaved. The non-sterile hand-help neuronavigation pointer was in turn used for surgical planning purposes in order to identify and digitally register in the neuronavigation system the optimal invasive lesion access trajectory that would reach the lesion without damaging the surrounding functional regions or white matter tracts. For example, in the case of a peri-central tumor, if the tumor presented with compressing effects displacing the motor and sensory tracts posteriorly, a posterior frontal trajectory would be most appropriate to approach the lesion. In the same case, if the tumor presented with compressing effects displacing the motor and sensory tracts anteriorly, an anterior parietal trajectory would be most appropriate to approach the lesion. If the tumor resided in between the primary motor and sensory cortices, displacing the motor fibers anteriorly and the sensory fibers posteriorly, a central sulcus entry approach to the tumor would be most appropriate.

Once the optimal invasive lesion access trajectory was determined, the craniotomy was planned, the patient’s scalp was sterilized and the surgical procedure would initiate. Craniotomies were limited to minimize the brain shift effect and were tailored with respect to the volume and the depth of the lesion. The brain shift was evaluated after the opening of the dura and exposure of the brain by placing a sterile hand-held neuronavigation probe on the brain surface and calculating the difference between the physical location and the surface location on MRI. This measurement was taken along the pre-planned trajectory of approach to the lesion, the brain shift interval was added as an offset value to the neuronavigation system and was in turn used to track the approach trajectory throughout the process. At regular intervals during the surgical process, the sterile hand-held neuronavigation pointer was used to verify compliance with the planned trajectory and confirm lesion resection margins with respect to the surrounding functional regions and tracts. In all and only in patients with motor symptomatology, additional intraoperative neurophysiological monitoring was used to complement and verify the functional neuroimaging modalities.

### 2.7. Outcomes and Statistics

Preoperative deficits and postoperative outcomes were evaluated after careful review of the patients’ medical records. Postoperative outcomes were evaluated in two follow-up clinic visits taking place approximately 1 month and 6 months following surgery. During the postoperative clinic visits, the patients underwent comprehensive neurological evaluation by typical means. The overall impression regarding the patient’s progress was a result of contrasting the preoperative neurological profile with the postoperative profile. For the purposes of this study, postoperative neurological status was classified into 5 distinct categories: preservation of asymptomatic—no deficit preoperative status, resolution of preoperative symptoms and signs, improvement of preoperative symptoms and/or signs, preservation of existing symptoms and/or neurological deficits and worsening of preoperative symptoms and/or neurological deficits.

The degree of association between categorical covariates representing the use of presurgical neuroimaging modalities and surgical outcome was assessed by a two-tailed Fisher’s exact test. The degree of association between categorical covariates, other than those representing the use of presurgical neuroimaging modalities, and surgical outcome was assessed by a two-sided Chi2. For both tests, the statistical threshold of *p* < 0.05 was used, and the tests were implemented in IBM SPSS Statistics 28.0.

## 3. Results

In our patient cohort, 115 patients were females (45.6%), mean age at the time of presurgical evaluation was 54 ± 16.5 years (range 5–85), 124 (49.2%) had left hemispheric lesions, 115 (45.6%) had right hemispheric lesions and 12 (4.7%) had lesions of bilateral hemispheric distribution. In terms of main lobar localization, 100 (39.6%) of them had frontal lesions (that may or may not include the insula), 56 (22.2%) had temporal lesions, 53 (21.0%) had parietal lesions, 23 (9.1%) had occipital lesions and 20 (7.9%) presented with lesions extending in two lobes. The majority of lesions were gliomas, and the overall distribution included Grade I abnormalities (69, 27.3%), such as meningiomas, cavernous angiomas, choroid plexus papillomas, colloid cysts, dermoid cysts, ependymomas, pilocytic astrocytomas and schwannomas; Grade II abnormalities (33, 13.0%), such as oligodentrogliomas, (fibrillary) astrocytomas, neurocytomas, meningiomas and oligostrocytomas; Grade III abnormalities (29, 11.5%), such as anaplastic oligoastrocytomas, anaplastic ependymomas, anaplastic oligodendrogliomas and anaplastic (pilocytic) astrocytomas; and Grade IV abnormalities (111, 44.0%), such as glioblastomas, gliosarcomas, chondrosarcomas and metastatic tumors. The rest of the non-graded intracerebral lesions (6, 2.3%) included abscess, encephalocele, histiocytosis, hydatid cyst, osteoma and hematoma. Preoperative neurological examination disclosed motor (68, 26.9%), somatic sensory (32, 12.6%), visual (23, 9.1%), mixed somatic sensory and motor (12, 4.7%), language and/or memory (8, 3.1%) deficits and other manifestations/findings such as psychotic and/or confusional/perceptual episodes, persistent headaches and intracranial hypertension (8, 3.1%). Epileptic seizures had presented in 47 patients, (18.6%) and 76 (30.1%) patients did not experience any preoperative symptoms and had a normal neurological examination.

By the end of the first postoperative month, in 54 patients (21.4%), all preoperative symptoms and deficits were resolved, 18 patients (7.14%) demonstrated improvement, 66 patients (26.1%) that had no preoperative symptoms remained symptom-free, 93 patients (36.9%) remained with the same preoperative deficits and 21 patients (8.3%) worsened. At 6 months postoperatively, the number of patients with completely resolved symptoms was increased (67, 26.5%), the number of patients that demonstrated improvement in their symptoms also increased (53, 21.0%), 69 patients (27.3%) were evaluated to be in their preoperative symptom-free state, the number of patients with the same preoperative deficits had decreased (50, 19.8%) and the number of patients with worsening had also decreased (13, 5.15%).

Hemispheric lateralization and distribution of the lesion had no significant effect on postoperative outcomes at 1 month (x^2^(2, 252) = 6.48, *p* = 0.59) and 6 months (x^2^(2, 252) = 6.19, *p* = 0.62). The lobar localization of the lesion did not present with significant effects on postsurgical outcomes at 1 month (x^2^(4, 252) = 17.04, *p* = 0.38) and 6 months (x^2^(4, 252) = 29.75, *p* = 0.19). Gender also had overall no effect on the emergence of postoperative outcomes at 1 month (x^2^(1, 252) = 1.09, *p* = 0.89) and 6 months (x^2^(1, 252) = 0.30, *p* = 0.98).

At 1 month postoperatively, patients in which neither DTI-tractography nor combined fMRI/DTI-tractography were used were less likely to present improvement in preoperative symptoms (P_GROUP_ = 0.005). In contrast, when DTI-tractography was used, patients were more likely to either present improvement or preservation in preoperative symptoms at 1 month postoperatively (*p* = 0.004 and *p* = 0.007, respectively). At 6 months follow-up, a significant number of patients for which combined fMRI/DTI-tractography was used for intraoperative neuronavigation showed complete resolution of their preoperative symptoms (P_GROUP_ < 0.001). Most patients that maintained their symptom-free condition postoperatively had neuronavigation with anatomical MRI only (P_GROUP_ < 0.001 for both 1 month and 6 months postoperatively) (Table 2).

Neuronavigation with anatomical MRI alone was the preferred approach in patients with low-grade I and II lesions (N_MRI_ = 68/102 vs. N_MRI/DTI_ = 18/102 vs N_MRI/DTI-fMRI_ = 16/102, *p* < 0.001), while no specific imaging protocol of choice was identified in high-grade III and IV lesions (N_MRI_ = 54/140 vs. N_MRI/DTI_ = 46/140 vs. N_MRI/DTI-fMRI_ = 40/140, *p* = 0.16). Among patients with low-grade lesions (N = 102), those that had combined fMRI/DTI-tractography were more likely to experience complete resolution of their preoperative symptoms at both 1 month and 6 month follow-ups (*p* = 0.001 and *p* < 0.001, respectively) (Table 3). Among those with high-grade lesions (N = 140), patients imaged with combined fMRI/DTI-tractography were more likely to have their preoperative symptoms resolved at 6 months after surgery (P_GROUP_ = 0.005). In the same high-grade lesion population, those more likely to maintain their symptom-free condition postoperatively were operated on with neuronavigation based on anatomical MRI only (P_GROUP_ = 0.001 for both 1 month and 6 month follow-ups) (Table 4).

Patients with motor symptoms were more likely to experience complete remission of preoperative symptoms at 6 months post-surgery when either DTI-tractography or combined fMRI/DTI-tractography was used (*p* = 0.008 and *p* = 0.004, respectively, P_GROUP_ = 0.005); although, there was no significant difference between the two imaging protocols (*p* = 1) (Table 5). Patients with sensory symptoms were more likely to experience complete remission of preoperative symptoms at 6 months post-surgery when combined fMRI/DTI-tractography was used (P_GROUP_ = 0.004) (Table 6). For patients with preoperative seizures, the presurgical neuroimaging approach used intraoperatively did not have any significant effect on their outcomes. The resolution of preoperative epileptic seizures was independent from any combination of neuroimaging modalities used during lesion resection under neuronavigation (Table 7).

## 4. Discussion

The purpose of this work was to assess the validity of complementing standard anatomical brain MRI with DTI-tractography alone or with combined fMRI/DTI-tractography in the neuronavigation process during brain surgery by means of patient outcome evaluation. DTI-tractography and fMRI, despite their widely acknowledged limitations [84,85,86], have dynamically entered the presurgical evaluation context of brain surgery during the past decades, providing novel perspectives in surgical planning and lesion access approaches [87,88,89]. Although the functional neuroimaging community is still in the process of managing their advantages and limitations to improve patient outcomes, both non-invasive modalities have been successfully used to determine the proximity of lesions to eloquent cortex, optimize the surgical lesion access trajectory and eventually tailor resections [71,90,91]. However, the acquisition and processing of both modalities introduce significant time and effort and increased costs [92,93], and thereby, the question of how to make the best use of them was open for us to approach [77,94,95].

To answer that question, we retrospectively reviewed the medical records of 252 consecutive patients admitted for brain surgery over a seven-year interval, during which structural and functional neuroimaging were progressively introduced in our clinical workflow. This progressive introduction of advanced neuroimaging modalities in the presurgical evaluation process resulted in a unique balanced dataset of patients treated with (N = 124) and without (N = 128) structural/functional neuroimaging modalities by the same neurosurgical team. As one of the main goals of brain tumor surgery is to relieve patients from associated risks and improve their quality of life [2,3,4,5,6,7,8,9,10,11], we focused on the confounding effect the surgical procedure can have in the patient’s symptoms at both the planning and intraoperative neuronavigation levels where the neuroimaging modalities play a crucial role. Therefore, we documented their neurological deficits before the operation, as well as at one- and six-month postoperative clinical follow-ups, and correlated patient outcomes with the neuroimaging modalities used during preoperative planning and intraoperative neuronavigation procedures. Our goal was to determine the combination of imaging modalities most appropriate for patients depending on lesion severity and preoperative symptoms experienced in order to make the presurgical process more efficient and improve the surgical planning and navigation procedures.

In the total volume of our cohort, both DTI-tractography and combined fMRI/DTI-tractography were associated with significantly improved outcomes at one month postoperatively. Furthermore, they were also associated with significant resolution of preoperative symptoms and deficits at six-month follow-up compared to the use of anatomical MRI sequences alone (Table 1). The assessment of postoperative risk assessed by means of fMRI has been shown to correlate up to 88% with positive clinical outcomes [43]. Our results are concordant with the published literature demonstrating significant effects of the presurgical and intraoperative use of these modalities on patient outcomes [44,45,66,67,68,90]. Postoperative motor PNDs were significantly reduced when DTI-tractography was used in the resection of peri-central tumors [96] and when combined fMRI/DTI-tractography was used in peri-insular tumors [97]. These modalities have shown particular added value when combined to delineate resection limits [97,98,99,100] and extend the patient’s median survival [96]. More specifically, it has been shown that the distance between the MRI-visible tumor and the fMRI BOLD cluster, as well as the distance between the MRI-visible tumor and the DTI-tractography-reconstructed tract, are both predictive of pre and postoperative deficits when the tumor is involving the motor network (primary motor cortex and corticospinal tract) and the expressive language network (Broca’s and the arcuate fasciculus) [97].

The added value of using structural and functional neuroimaging was also highlighted when we clustered our patients based on lesion severity. In patients with low-grade (I and II) lesions, our results demonstrated that the combination of fMRI/DTI-tractography is significantly associated with resolution of preoperative symptoms at both postoperative follow-up intervals studied in this work. In patients with high-grade (III and IV) lesions, we found that the employment of both advanced neuroimaging modalities is associated with preoperative symptom remission at six months postoperatively (Table 2). A similar lag in postoperative improvement between the two lesion severity groups has been reported before [100], although that study used only DTI-tractography vs. anatomical MRI. Prior investigations have already shown that combined fMRI/DTI-tractography is superior to fMRI alone in evaluating the risk of resection proximity to eloquent structures and consequently in establishing improved patient outcomes [90,95,101]. Before the introduction of functional neuroimaging in the neurosurgical brain tumor resection workflows, postoperative neurological complications ranged from 15 to 33% [102,103,104]. With the refinements in resective planning introduced by functional neuroimaging modalities, the postoperative complication percentages have dropped to less than half [100,101]. Our results support and complement these studies, as they demonstrate the superiority of combined advanced neuroimaging modalities against anatomical MRI alone and anatomical MRI coupled with DTI-tractography and, thus, strongly support their use in presurgical planning and intraoperative neuronavigation of patients with preoperative symptoms.

In our full cohort analysis, patients without any preoperative symptoms maintained their symptom-free condition at both one- and six-month follow-ups when operated on with anatomical MRI alone (Table 1). At first glance, this could represent a sampling bias introduced by the fact that most patients with low-grade I and II lesions, the majority of which do not present with major symptomatology [105], were offered presurgical imaging with anatomical MRI alone for planning and neuronavigation. However, when patient groups were subcategorized in terms of lesion severity, our analysis showed that these patients also belonged to the high-grade (III and IV) lesion group (Table 2). The lack of persistent presurgical symptoms in lesions of such severity is an indication that the lesion has not infiltrated and/or has not been pressing eloquent cortical areas, which typically results in functional deficits, or cortical regions in general, which typically can generate epileptic seizures [106,107]. Our results, therefore, suggest that the specific category of patients can be safely treated without advanced neuroimaging datasets supporting the surgical procedures.

We further investigated whether advanced neuroimaging modalities have an effect on particular preoperative findings and clustered three subgroups of patients demonstrating the main categories of presurgical symptoms registered: motor deficits, sensory deficits and epileptic seizures. In the motor deficit group, a significant increase in patients with complete resolution of preoperative symptoms at six months was associated with the use of either DTI-tractography or combined fMRI/DTI-tractography, without demonstrating a significant difference between them. This result supports the use of any advanced neuroimaging modality when the patient presents with motor symptoms in order to maximize the surgical benefit for the patient. However, the lack of significant difference among the two neuroimaging protocols suggests that the use of fMRI in such patients may be redundant. Our interpretation of this result is that for the delineation of the primary motor regions, DTI-tractography can provide a more complete map of their extent across the pre-central gyrus [29,108] compared to fMRI due to the limitations of the latter imposed by task-specific activations [77,84,94].

In the sensory deficit group, the significant increase in patients with complete resolution of preoperative symptoms at six months was associated with the use of combined fMRI/DTI-tractography. Although this result clearly favors the use of combined fMRI/DTI-tractography for the optimal surgical planning of these patients, DTI-tractography alone does not appear as advantageous as in the motor deficit group, even though DTI-tractography provides superior resolution in mapping the primary somatosensory regions compared to fMRI, which suffers from task-related limitations. As the sensory group included patients with disturbances of somatic sensation and vision, it seems that the introduction of fMRI provides added value to the delineation of the primary and associative visual areas [37,80,109], the anatomical preservation of which highly contributes to a favorable surgical outcome [110].

Finally, the epileptic seizure group demonstrates that the use of advanced neuroimaging modalities may not provide clear advantages to the postoperative improvement and/or resolution of preoperative seizures. A similar effect has been shown before when investigating the added value of intraoperative MRI [96]. Our explanation for this effect is that the manifestation of seizures in the brain tumor setting is not associated with a particular functional network in the brain, therefore tumor resection with or without functional neuroimaging modalities can have the same beneficial outcome.

An important feature deriving from our analysis was the lack of worsening symptoms at the six-month follow-up in patients with neuronavigation facilitated by both DTI-tractography and fMRI (Table 1). A similar feature was derived from the preoperative motor and sensory symptoms subgroups, where the use of either imaging protocol also resulted in zero patients with worsening symptoms at six months postoperatively. Specifically for the sensory group, this was the case as early as the first postoperative month (Table 3). Although these traits are hard to be depicted by descriptive statistics, they may be of major importance when decisions are made regarding the appropriate preoperative dataset and when the postoperative risks for the patient are estimated [111,112]. These outcomes demonstrate that the use of advanced mapping modalities may protect patients undergoing brain surgery from progression of symptoms and may benefit the patient’s postoperative clinical profile.

Our study focused on the effect of presurgical and intraoperative advanced neuroimaging on patient outcomes and is, thereby, limited in terms of more quantitative measures, such as proximity metrics of tracts and eloquent regions to the target lesions, volumetric data to evaluate the extent of resection (gross-total resection vs. subtotal resection), etc. However, the fact that all resective procedures and decisions were performed by a single neurosurgeon, and that all presurgical structural and functional mapping was following a consistent strategy, as described in detail in the Section 2, suggests a high degree of uniformity across cases. Another objective limitation of our study regards the short postoperative follow-up interval of six months, which was set due to the high morbidity rate of high-grade lesions. Another set of limitations derives from the methods used in our study. More specifically, the inability to feed fMRI data as ROIs for the fiber-tracking algorithm, due to the fact that they were processed separately and independently, may have significantly underestimated the potential of the combined fMRI/DTI-tractography approach. In addition, the lack of broadly adopted fMRI task protocols renders replication of our results challenging. In this study, we did not differentiate tumors on the basis of being intra- versus extra-parenchymal, and the reason for doing that is two-fold: 1. At the time of introducing functional neuroimaging in our surgical workflow, there were no widely acknowledged criteria of patient inclusion–exclusion, and we applied our functional neuroimaging modalities in case of large extra-parenchymal tumors residing in the immediate vicinity or applying notable pressure on eloquent cortical regions; this approach becomes particularly important given the growing literature demonstrating infiltrative forms of extra-tentorial tumors [113,114,115]. 2. There were indications from the published literature that parenchymal distortions caused by extra-tentorial lesions can be depicted by DTI-tractography as respective distortions in underlying fiber tracts [116], which in turn can manifest as PNDs. Including these tumors in our study remained in accordance with our main goal, which was the evaluation of the functional neuroimaging methods. Another limitation of our study derives from the fact that our center expertise is confined to brain tumor surgery, and patients are referred to external specialized centers for chemotherapy and radiation therapy. This workflow does not allow for a complete follow-up of our patients, therefore complete survival data were not available for our study. It is also worth mentioning that the development of more sophisticated approaches, such as constrained spherical deconvolution [117,118,119] and generalized q-sampling imaging [65,120], may reduce the current methodological limitations of DTI-tractography. Respectively, the constraints imposed by the variability of fMRI tasks [77] may be also significantly reduced by the introduction of resting-state fMRI approaches [121,122,123].

## 5. Conclusions

The individual or combined use of DTI-tractography and fMRI has repeatedly been shown to provide added value and increase confidence in resective brain lesion procedures, thereby supporting a high level of postoperative quality of life for the patients. However, their use has not been uniform or standardized and still lacks predictive factors [97,124,125]. In that context, our study attempted to partly fill these gaps, 1. by outlining a lesion-specific mapping strategy, depending on the lesion’s location and extent, and 2. by associating the use of advanced neuroimaging modalities with surgical outcomes, in a unique cohort that included a control subgroup. At one month postoperatively, DTI-tractography patients were more likely to present either improvement or preservation of PNDs. At six months, combined DTI-tractography/fMRI patients were more likely to experience complete PND resolution. Low-grade lesion patients with combined DTI-tractography/fMRI were more likely to experience complete resolution of PNDs at one and six months. High-grade lesion patients with combined DTI-tractography/fMRI were more likely to have PNDs resolved at six months. Patients with motor symptoms were more likely to experience complete remission of PNDs at six months with DTI-tractography or combined DTI-tractography/fMRI, without significant difference between the two imaging protocols. Patients with sensory symptoms were more likely to experience complete PND remission at six months with combined DTI-tractography/fMRI. The intraoperative neuroimaging modality did not have a significant effect in patients with preoperative seizures. Lack of PND worsening was observed at six-month follow-up in patients with combined DTI-tractography/fMRI. Our results strongly support the combined use of DTI-tractography and fMRI in patients undergoing brain surgery for the purpose of maintaining and/or improving their postoperative clinical profile. Our work also provides insights on patient profiles that would benefit the most from functional neuroimaging workup, as well as on those for which such processes could be redundant.

## Figures and Tables

**Figure 1 brainsci-13-01574-f001:**
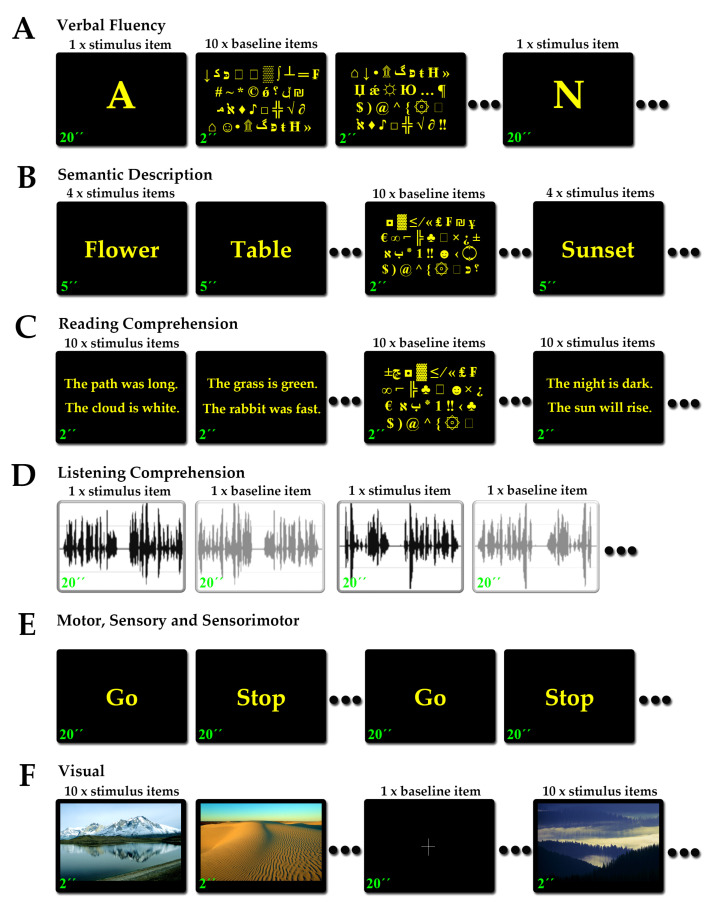
Flowcharts of the fMRI tasks. (**A**) Verbal fluency task: patients were presented with a single letter and were instructed to silently generate words starting with that letter. (**B**) Semantic description task: patients were presented with a single word of objects and tools of household and urban life and were instructed to silently generate a complete sentence with a definition for this object or a description of its common use. (**C**) Reading comprehension task: patients were presented with sentences of brief factual statements and were instructed to read them silently as soon as they appeared on screen. (**D**) Listening comprehension task: patients were asked to listen carefully to a pre-recorded narrative and be prepared to answer questions on the content. (**E**) Motor, sensory and sensorimotor tasks: the patients were presented with the simple instructions “Go” to begin the activity and “Stop” to stop the activity. During the sensory task, “Go” and “Stop” were instructions for the functional neuroimager who was performing somatosensory stimulation. (**F**) Visual task: the patient was instructed to visually explore the complex colorful images. Durations for each item appear in green on the left lower corner of each item; this was not part of the presentation to the patients.

**Figure 2 brainsci-13-01574-f002:**
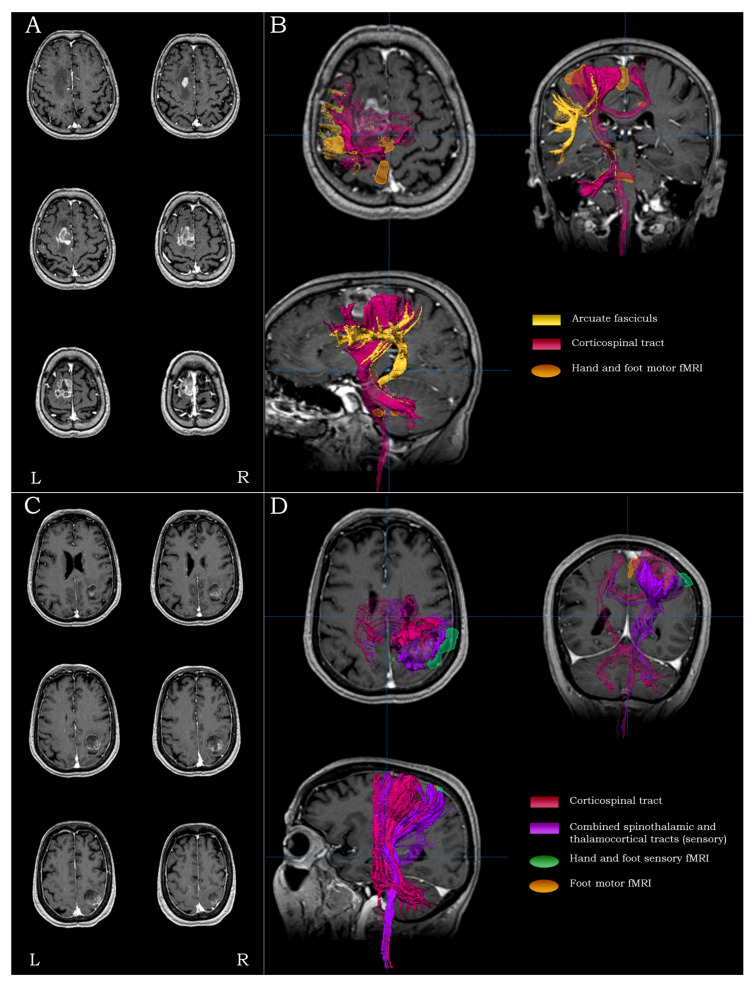
Examples highlighting the strategy and rationale of functional and structural peri-lesional coverage. Anatomical localization and morphological depiction (**A**), along with fMRI and DTI-tractography mapping (**B**), on post-G Gadolinium T1-weighted images for a precentral tumor. Anatomical localization and morphological depiction (**C**), along with fMRI and DTI-tractography mapping (**D**), on post-Gadolinium T1-weighted images for a central tumor.

**Figure 3 brainsci-13-01574-f003:**
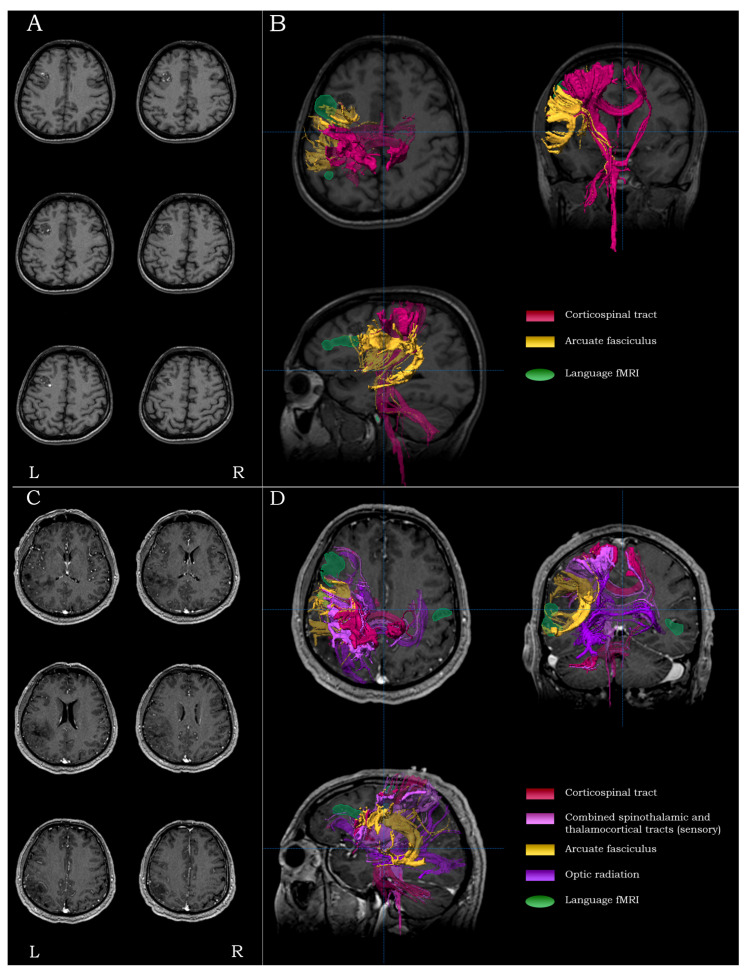
Examples highlighting the strategy and rationale of functional and structural peri-lesional coverage. Anatomical localization and morphological depiction (**A**), along with fMRI and DTI-tractography mapping (**B**), on post-G Gadolinium T1-weighted images for an inferior frontal tumor. Anatomical localization and morphological depiction (**C**), along with fMRI and DTI-tractography mapping (**D**), on post-G Gadolinium T1-weighted images for an inferior parietal tumor.

**Figure 4 brainsci-13-01574-f004:**
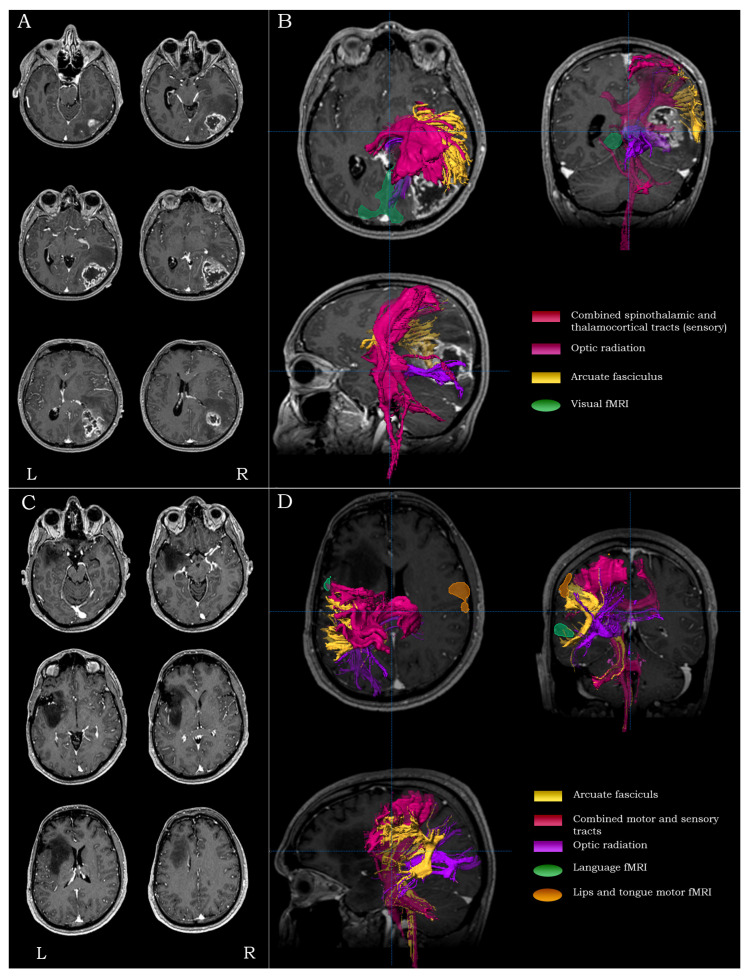
Examples highlighting the strategy and rationale of functional and structural peri-lesional coverage. Anatomical localization and morphological depiction (**A**), along with fMRI and DTI-tractography mapping (**B**), on post-G Gadolinium T1-weighted images for a posterior quadrant tumor. Anatomical localization and morphological depiction (**C**), along with fMRI and DTI-tractography mapping (**D**), on post-G Gadolinium T1-weighted images for a multi-lobar (frontal/insular/temporal) tumor.

**Table 1 brainsci-13-01574-t001:** Region-specific surgical planning coverage strategy using fMRI and DTI-tractography. Asterisk (*) marks optional modalities depending on the extent of the lesion.

Lesion Location	fMRI				DTI-Tractography			
	Language	Motor	Sensory	Sensorimotor	Motor Pathway	Sensory Pathway	Arcuate Fasciculus	Optic Radiation
Anterior frontal	√ *						√	
Inferior frontal	√				√		√	
Pre-central	√ *	√			√		√	
Central		√	√		√	√		
Superior parietal	√		√ *			√	√ *	
Inferior parietal				√	√	√	√	√ *
Occipital			√					√
Posterior temporal	√			√	√	√	√ *	√
Anterior temporal								√

**Table 2 brainsci-13-01574-t002:** The use of neuroimaging modalities during neuronavigation vs. surgical outcomes in the patient population of this study. P_MRIvsMRI/DTI_, P_MRIvsMRI/DTI-fMRI_, P_MRI/DTIvs MRI/DTI-fMRI_ represent 2 × 2 Fisher’s Exact tests between the respective neuroimaging modalities. P_GROUP_ represents a 3 × 2 Fisher exact test among all neuroimaging modalities. Statistically significant values are presented in bold/italic fonts followed by an asterisk (PNDs = preoperative neurological deficits; DTI = diffusion tensor imaging; fMRI = functional magnetic resonance imaging).

	Neuroimaging Modalities Used During Neuronavigation			Statistics (Fisher’s Exact Test)			
Patient Population = 252	Anatomical MRI (N = 128)	Anatomical MRI + DTI-Tractography (N = 68)	Anatomical MRI + DTI-Tractography + fMRI (N = 56)	P_MRIvsMRI/DTI_	P_MRIvsMRI/DTI-fMRI_	P_MRI/DTIvs MRI/DTI-fMRI_	P_GROUP_
Outcomes at 1 month postoperative (N; %)							
Resolution of PNDs	24; 18.8	10; 14.7	20; 35.7	0.552	0.015	0.011	0.011
Improvement of PNDs	3; 2.3	9; 13.2	6; 10.7	** *0.004 ** **	0.024	0.785	** *0.005 ** **
Preservation of asymptomatic status	52; 40.6	9; 13.2	5; 8.9	** *<0.001 ** **	** *<0.001 ** **	0.572	** *<0.001 ** **
Preservation of PNDs	38; 29.6	34; 50.0	21; 37.5	** *0.007 ** **	0.307	0.204	0.019
Worsening of PNDs	11; 8.5	6; 8.8	4; 7.1	1	1	1	0.999
Outcomes at 6 months postoperative (N; %)							
Resolution of PNDs	24; 18.7	16; 23.5	27; 48.2	0.459	** *<0.001 ** **	** *0.004 ** **	** *<0.001 ** **
Improvement of PNDs	21; 16.4	22; 32.3	10; 17.8	0.017	0.832	0.098	0.034
Preservation of asymptomatic status	52; 40.6	9; 13.2	8; 14.2	** *<0.001 ** **	** *<0.001 ** **	1	** *<0.001 ** **
Preservation of PNDs	21; 16.4	18; 26.4	11; 19.6	0.131	0.673	0.401	0.246
Worsening of PNDs	10; 7.8	3; 4.4%	0; 0.0	0.548	0.033	0.251	0.069

**Table 3 brainsci-13-01574-t003:** The use of neuroimaging modalities during neuronavigation vs. surgical outcomes in the subgroup of our patient population diagnosed with benign (Grade I and II) brain tumors. P_MRIvsMRI/DTI_, P_MRIvsMRI/DTI-fMRI_, P_MRI/DTIvsMRI/DTI-fMRI_ represent 2 × 2 Fisher’s Exact tests between the respective neuroimaging modalities. P_GROUP_ represents a 3 × 2 Fisher’s Exact test among all neuroimaging modalities. Statistically significant values are presented in bold/italic fonts followed by an asterisk (PNDs = preoperative neurological deficits; DTI = diffusion tensor imaging; fMRI = functional magnetic resonance imaging).

	Neuroimaging Modalities Used During Neuronavigation			Statistics (Fisher’s Exact Test)			
Grade I and II Lesions Patient Population = 102	Anatomical MRI (N = 68)	Anatomical MRI + DTI-Tractography (N = 18)	Anatomical MRI + DTI-Tractography + fMRI (N = 16)	P_MRIvsMRI/DTI_	P_MRIvsMRI/DTI-fMRI_	P_MRI/DTIvs MRI/DTI-fMRI_	P_GROUP_
Outcomes at 1 month postoperative (N; %)							
Resolution of PNDs	15; 22.0	5; 27.7	11; 68.7	0.754	** *<0.001 ** **	0.003	** *0.001 ** **
Improvement of PNDs	1; 1.4	2; 11.1	3; 18.7	0.109	0.020	0.648	0.013
Preservation of asymptomatic status	33; 48.5	6; 33.3	2; 12.5	0.295	0.011	0.405	0.020
Preservation of PNDs	14; 20.5	4; 22.2	0; 0.0	1	0.061	0.105	0.114
Worsening of PNDs	5; 7.3	1; 5.5	0; 0.0	1	0.577	1	0.825
Outcomes at 6 months postoperative (N; %)							
Resolution of PNDs	15; 22.0	7; 38.8	14; 87.5	0.222	** *<0.001 ** **	** *0.005 ** **	** *<0.001 ** **
Improvement of PNDs	10; 14.7	4; 22.2	0; 0.0	0.478	0.195	0.105	0.160
Preservation of asymptomatic status	33; 48.5	6; 33.3	2; 12.5	0.295	0.011	0.232	0.020
Preservation of PNDs	5; 7.3	0; 0.0	0; 0.0	0.579	0.577	1	0.494
Worsening of PNDs	5; 7.3	1; 5.5	0; 0.0	1	0.577	1	0.830

**Table 4 brainsci-13-01574-t004:** The use of neuroimaging modalities during neuronavigation vs. surgical outcomes in the subgroup of our patient population diagnosed with malignant (Grade III and IV) brain tumors. P_MRIvsMRI/DTI_, P_MRIvsMRI/DTI-fMRI_, P_MRI/DTIvsMRI/DTI-fMRI_ represent 2 × 2 Fisher’s Exact tests between the respective neuroimaging modalities. P_GROUP_ represents a 3 × 2 Fisher’s Exact test among all neuroimaging modalities. Statistically significant values are presented in bold/italic fonts followed by an asterisk (PNDs = preoperative neurological deficits; DTI = diffusion tensor imaging; fMRI = functional magnetic resonance imaging).

	Neuroimaging Modalities Used During Neuronavigation			Statistics (Fisher’s Exact Test)			
Grade III and IV Lesions Patient Population = 140	Anatomical MRI (N = 54)	Anatomical MRI + DTI-Tractography (N = 46)	Anatomical MRI + DTI-Tractography + fMRI (N = 40)	P_MRIvsMRI/DTI_	P_MRIvsMRI/DTI-fMRI_	P_MRI/DTIvs MRI/DTI-fMRI_	P_GROUP_
Outcomes at 1 month postoperative (N; %)							
Resolution of PNDs	8; 14.8	5; 10.8	9; 22.5	0.766	0.419	0.393	0.343
Improvement of PNDs	2; 3.7	5; 10.8	3; 7.5	0.242	0.647	0.718	0.387
Preservation of asymptomatic status	15; 27.7	2; 4.3	3; 7.5	** *0.002 ** **	0.016	0.660	** *0.001 ** **
Preservation of PNDs	23; 42.5	30; 65.2	21; 52.5	0.028	0.405	0.274	0.085
Worsening of PNDs	6; 11.1	4; 8.6	4; 0.1	0.749	1	1	0.939
Outcomes at 6 months postoperative (N; %)							
Resolution of PNDs	8; 14.8	8; 17.3	17; 42.5	0.788	** *0.004 ** **	0.016	** *0.005 ** **
Improvement of PNDs	10; 18.5	17; 36.9	9; 22.5	0.044	0.795	0.165	0.933
Preservation of asymptomatic status	15; 27.7	2; 4.3	3; 7.5	** *0.002 ** **	0.016	0.655	** *0.001 ** **
Preservation of PNDs	16; 29.6	18; 39.1	11; 27.5	0.397	1	0.360	0.483
Worsening of PNDs	5; 9.2	1; 2.1	0; 0.0	0.213	0.069	1	0.091

**Table 5 brainsci-13-01574-t005:** The use of neuroimaging modalities during neuronavigation vs. surgical outcomes in the subgroup of our patient population that presented with preoperative motor deficits. P_MRIvsMRI/DTI_, P_MRIvsMRI/DTI-fMRI_, P_MRI/DTIvsMRI/DTI-fMRI_ represent 2 × 2 Fisher’s Exact tests between the respective neuroimaging modalities. P_GROUP_ represents a 3 × 2 Fisher’s Exact test among all neuroimaging modalities. Statistically significant values are presented in bold/italic fonts followed by an asterisk (PNDs = preoperative neurological deficits; DTI = diffusion tensor imaging; fMRI = functional magnetic resonance imaging).

	Neuroimaging Modalities Used During Neuronavigation			Statistics (Fisher’s Exact Test)			
Preoperative Motor Symptoms Patient Population = 80	Anatomical MRI (N = 32)	Anatomical MRI + DTI-Tractography (N = 27)	Anatomical MRI + DTI-Tractography + fMRI (N = 21)	P_MRIvsMRI/DTI_	P_MRIvsMRI/DTI-fMRI_	P_MRI/DTIvs MRI/DTI-fMRI_	P_GROUP_
Outcomes at 1 month postoperative (N; %)							
Resolution of PNDs	1; 3.1	0; 0.0	0; 0.0	1	1	1	0.999
Improvement of PNDs	1; 3.1	7; 25.9	5; 23.8	0.018	0.030	1	0.024
Preservation of PNDs	27; 84.3	17; 62.9	15; 71.4	0.076	0.310	0.758	0.093
Worsening of PNDs	3; 9.3	3; 11.1	1; 4.7	1	1	0.621	0.884
Outcomes at 6 months postoperative (N, %)							
Resolution of PNDs	1; 3.1	8; 29.6	7; 33.3	** *0.008 ** **	** *0.004 ** **	1	** *0.005 ** **
Improvement of PNDs	16; 50.0	10; 37.0	8; 38.0	0.430	0.416	1	0.659
Preservation of PNDs	13; 40.6	8; 29.6	6; 28.5	0.424	0.399	1	0.605
Worsening of PNDs	2; 6.2	0; 0.0	0; 0.0	0.495	0.512	1	0.334

**Table 6 brainsci-13-01574-t006:** The use of neuroimaging modalities during neuronavigation vs. surgical outcomes in the subgroup of our patient population that presented with preoperative sensory deficits. P_MRIvsMRI/DTI_, P_MRIvsMRI/DTI-fMRI_, P_MRI/DTIvsMRI/DTI-fMRI_ represent 2 × 2 Fisher’s Exact tests between the respective neuroimaging modalities. P_GROUP_ represents a 3 × 2 Fisher’s Exact test among all neuroimaging modalities. Statistically significant values are presented in bold/italic fonts followed by an asterisk (PNDs = preoperative neurological deficits; DTI = diffusion tensor imaging; fMRI = functional magnetic resonance imaging).

	Neuroimaging Modalities Used During Neuronavigation			Statistics (Fisher’s Exact Test)			
Preoperative Sensory SymptomsPatient Population = 44	Anatomical MRI (N = 15)	Anatomical MRI + DTI-Tractography (N = 21)	Anatomical MRI + DTI-Tractography + fMRI (N = 8)	P_MRIvsMRI/DTI_	P_MRIvsMRI/DTI-fMRI_	P_MRI/DTIvs MRI/DTI-fMRI_	P_GROUP_
Outcomes at 1 month postoperative (N; %)							
Resolution of PNDs	0; 0.0	0; 0.0	0; 0.0	1	1	1	1
Improvement of PNDs	1; 6.6	3; 14.2	2; 25.0	0.625	0.268	0.596	0.369
Preservation of PNDs	11; 73.3	18; 85.7	6; 75.0	0.417	1	0.596	0.611
Worsening of PNDs	3; 20.0	0; 0.0	0; 0.0	0.063	0.525	1	0.070
Outcomes at 6 months postoperative (N, %)							
Resolution of PNDs	0; 0.0	2; 9.5	4; 50.0	0.500	** *0.007 ** **	0.033	** *0.004 ** **
Improvement of PNDs	5; 33.3	9; 42.8	1; 12.5	0.731	0.369	0.200	0.315
Preservation of PNDs	7; 46.6	10; 47.6	3; 37.5	1	1	0.696	0.924
Worsening of PNDs	3; 20.0	0; 0.0	0; 0.0	0.063	0.525	1	0.070

**Table 7 brainsci-13-01574-t007:** The use of neuroimaging modalities during neuronavigation vs. surgical outcomes in the subgroup of our patient population that presented with preoperative epileptic seizures. P_MRIvsMRI/DTI_, P_MRIvsMRI/DTI-fMRI_, P_MRI/DTIvsMRI/DTI-fMRI_ represent 2 × 2 Fisher’s Exact tests between the respective neuroimaging modalities. P_GROUP_ represents a 3 × 2 Fisher’s Exact test among all neuroimaging modalities. PNDs = preoperative neurological deficits; DTI = diffusion tensor imaging; fMRI = functional magnetic resonance imaging.

	Neuroimaging Modalities Used During Neuronavigation			Statistics (Fisher’s Exact Test)			
Preoperative Seizures Patient Population = 47	Anatomical MRI (N = 21)	Anatomical MRI + DTI-Tractography (N = 10)	Anatomical MRI + DTI-Tractography + fMRI (N = 16)	P_MRIvsMRI/DTI_	P_MRIvsMRI/DTI-fMRI_	P_MRI/DTIvs MRI/DTI-fMRI_	P_GROUP_
Outcomes at 1 month postoperative (N; %)							
Resolution of PNDs	19; 90.4	10; 100	16; 100	0.548	0.495	1	0.689
Improvement of PNDs	0; 0.0	0; 0.0	0; 0.0	1	1	1	1
Preservation of PNDs	1; 4.76	0; 0.0	0; 0.0	1	1	1	0.999
Worsening of PNDs	1; 4.76	0; 0.0	0; 0.0	1	1	1	0.999
Outcomes at 6 months postoperative (N, %)							
Resolution of PNDs	19; 90.4	10; 100	16; 100	0.548	0.495	1	0.689
Improvement of PNDs	1; 4.76	0; 0.0	0; 0.0	1	1	1	0.999
Preservation of PNDs	0; 0.0	0; 0.0	0; 0.0	1	1	1	1
Worsening of PNDs	1; 4.76	0; 0.0	0; 0.0	1	1	1	0.999

## Data Availability

All data are available by the corresponding author upon request. The data are not publicly available.

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
