# Peer review of "Functional Magnetic Resonance Imaging and Diffusion Tensor Imaging-Tractography in Resective Brain Surgery: Lesion Coverage Strategies and Patient Outcomes"

_brainsci, 2023, doi:10.3390/brainsci13111574_

Round 1
Reviewer 1 Report
Comments and Suggestions for Authors
Authors evaluate diffusion tensor imaging (DTI)-tractography and combined DTI-tractography with functional magnetic resonance imaging (fMRI) during intra-operative neuronavigation in resective brain surgery, using lesion-related preoperative neurological deficit (PND) outcomes as metrics. They retrospectively reviewed 252 patients admitted for brain surgery. They underwent neuroimaging pre-operative evaluation with conventional anatomical neuroimaging protocols (n=127), additional DTI-tractography (n=69), and combined DTI-tractography/fMRI(n = 56). One month and six months postoperative evaluation for symptoms showed the combined use of DTI-tractography and fMRI in patients undergoing respective brain surgery for improving their postoperative quality of life. This study indicates the importance of appropriate preoperative neuroimaging evaluation. However, there are a few points they should address.
The authors performed various tasks for fMRI. It would be better to have a figure to show all kinds of tasks and procedures (how many blocks they had and how long the task took.)
I'm not sure if all participants (n = 56) underwent all fMRI tasks, or if they picked some for them.
I would like to know the reasons why the authors selected those fMRI tasks. Are there any previous reports that showed these tasks could elicit neural activity in selected ROIs?
There was no detailed description of image processing. I think they can present this in the supplementary file, but still, they need to describe how to process fMRI and DTI data and thresholds for the 1st-level analysis.
It would be better to show the table about the demographics of patients, their disorders that needed surgery, and the severity of those disorders.
I'm curious if there is an effect of age, sex, or duration of illness on surgical outcomes or the associations between preoperative neuroimaging and surgical outcomes.
I think the authors well described the associations between preoperative neuroimaging exams and outcomes. But, to me, it wasn't clear what kind of preoperative neuroimaging is optimal for which location of legion or what grade of lesion. Could you describe the summary of statistics in somewhere the conclusion?
Minor points
I wasn't sure if patients with severe symptoms could perform fMRI evaluation. Are there any criteria for fMRI preoperative exam?
Reviewer 2 Report
Comments and Suggestions for Authors
Please revise the abstract and insert more information about the background and the method section.
Please better explain fMRI task. More information about language (i.e., are language task including explicit language production?) and motor task are needed to describe the task. Please describe how stimulus were provided to the patients, the instruction that patients received and if they performed a simulation out of the scanner. Please better explain how the off condition was constructed. See Ciavarro et al. 2021 for more information on the importance of an accurate off condition in fMRI.
Please better describe why the authors choose to reconstruct only 4 main tracts. Other tracts play a role in language functions.
Please describe which measures have been used to assess patients preoperative and postoperative conditions.
To perform a follow-up study including both intratentorial and extratentorial lesions needs a clear rationale or to treat those patients separately in the analysis. Please address how the authors solve this issue. Importantly, report the percentage of patients in which neuroimaging (DTI and fMRI) was used in preoperative planning for each istological group. Moreover, report the percentage of recovery for each istological group, and present differences between group treated with preoeperative planning and not preoperative planning.
Comments on the Quality of English LanguageLanguage quality is adequate
Reviewer 3 Report
Comments and Suggestions for Authors
The authors had a good sample size and used refined method to evaluate DTI and fMRI+DTI in patients with resective brain surgery. However, the discussion is poor and needs major imporvement. In the introduction, authors tend to switch from topic to topic and it lacks flow. It needs to be reframed.
The mention of the word quality of life in the manuscript is misleading. The authors did not actually measure the "quality of life" with psychophysical testing post-operative.
Are translations of verbal fluency task standardized? Also, in line 133, listening comprehension was only performed in Greek, or translations were provided in maternal languages as well. And was the meaning intact poss translations?
In the methods, reframe section 2.3.3 somatosensory tasks. It is unclear. Provide more information about the method with proper citation.
Section 2.4. Provide references to the masks used.
Section 2.4. was it a deterministic or probabilistic DTI? Because at least 18 durectiosn are needed to perform probabilistic tractography.
Line 380, statistical threshold of p < 0.01 (?). Is it a typo?
In the results section, line number 424, 412 seems confusing, what do the authors mean by at 6 months postoperatively...... resolved increased? and also... improvement also increased?
To replicate the methodolgy a few lines for the softwares used for fMRI and DTI analysis should be specified in the main manuscript.
Discussion should always start with the findings of the current study and then it is backed up with the previous literature and limitations. The discussion is better framed than the introduction. I suggest revising the manuscript.
Methodological limitations have not been mentioned in the discussion.
Comments on the Quality of English Language
The manuscript needs to be proofread by a native English speaker. There are many typos and grammatical errors which makes it extremely difficult for the reader to understand.
Round 2
Reviewer 1 Report
Comments and Suggestions for Authors
I think the revised manuscript is adequate for publication.
Author Response
Thank you for your productive comments, allowing us to improve our paper.
Reviewer 2 Report
Comments and Suggestions for Authors
Thank you for the revisions.
Some other issues need to be addressed.
To state the importance of neuroimaging in neurosurgery management you have to prove the role of neuroimaging in major resections. Please, insert data on lesion entity of resections and investigate whether neuroimaging has a role in performing gross total resections.
Minor issue: please add information on patients' overall survival, particularly important in high-grade lesions.
Author Response
See our response in the attached file.

Reviewer 3 Report
Comments and Suggestions for Authors
Thank you for revising the manuscript. It has been majorly improved.
Author Response

(The authors gave the same response as above.)
